# Structural Health Monitoring of Dams Based on Acoustic Monitoring, Deep Neural Networks, Fuzzy Logic and a CUSUM Control Algorithm

**DOI:** 10.3390/s22072482

**Published:** 2022-03-24

**Authors:** Luan Carlos de Sena Monteiro Ozelim, Lucas Parreira de Faria Borges, André Luís Brasil Cavalcante, Enzo Aldo Cunha Albuquerque, Mariana dos Santos Diniz, Manuelle Santos Góis, Katherin Rocio Cano Bezerra da Costa, Patrícia Figuereido de Sousa, Ana Paola do Nascimento Dantas, Rafael Mendes Jorge, Gabriela Rodrigues Moreira, Matheus Lima de Barros, Fernando Rodrigo de Aquino

**Affiliations:** Department of Civil and Environmental Engineering, University of Brasilia, Brasilia 70910-900, Brazil; lucaspdfborges@gmail.com (L.P.d.F.B.); abrasil@unb.br (A.L.B.C.); enzo.aldo@aluno.unb.br (E.A.C.A.); diniz.santos@aluno.unb.br (M.d.S.D.); manuellegeo@unb.br (M.S.G.); katherin.cano@aluno.unb.br (K.R.C.B.d.C.); figuereido.patricia@aluno.unb.br (P.F.d.S.); ana.paola@aluno.unb.br (A.P.d.N.D.); jorge.rafael@aluno.unb.br (R.M.J.); moreira.gabriela@aluno.unb.br (G.R.M.); barros.lima@aluno.unb.br (M.L.d.B.); fernando.aquino@aluno.unb.br (F.R.d.A.)

**Keywords:** structural monitoring, dams, geotechnical engineering, deep learning, autoencoder, fuzzy logic, CUSUM algorithm

## Abstract

Internal erosion is the most important failure mechanism of earth and rockfill dams. Since this type of erosion develops internally and silently, methodologies of data acquisition and processing for dam monitoring are crucial to guarantee a safe operation during the lifespan of these structures. In this context, artificial intelligence techniques show up as tools that can simplify the analysis and verification process not of the internal erosion itself, but of the effects that this pathology causes in the response of the dam to external stimuli. Therefore, within the scope of this paper, a methodological framework for monitoring internal erosion in the body of earth and rockfill dams will be proposed. For that, artificial intelligence methods, especially deep neural autoencoders, will be used to treat the acoustic data collected by geophones installed on a dam. The sensor data is processed to identify patterns and anomalies as well as to classify the dam’s structural health status. In short, the acoustic dataset is preprocessed to reduce its dimensionality. In this process, for each second of acquired data, three parameters are calculated (Hjorth parameters). For each parameter, the data from all the available sensors are used to calibrate an autoencoder. Then, the reconstruction error of each autoencoder is used to monitor how far from the original (normal) state the acoustic signature of the dam is. The time series of reconstruction errors are combined with a cumulative sum (CUSUM) algorithm, which indicates changes in the sequential data collected. Additionally, the outputs of the CUSUM algorithms are treated by a fuzzy logic framework to predict the status of the structure. A scale model is built and monitored to check the effectiveness of the methodology hereby developed, showing that the existence of anomalies is promptly detected by the algorithm. The framework introduced in the present paper aims to detect internal erosion inside dams by combining different techniques in a novel context and methodological workflow. Therefore, this paper seeks to close gaps in prior studies, which mostly treated just parts of the data acquisition–processing workflow.

## 1. Introduction

Until the early 20th century, notable structures were essentially bridges, tunnels, aqueducts, and canals, mostly built of wood and metal. In general, their maintenance process was carried out by specific agencies, which took care of the structures sectorially, and in a very Cartesian manner. Typically, the workers performing maintenance were responsible for a small portion of the structure and had perfect knowledge about the few units they were responsible for maintaining. This way, the notions of risk and damage to people and property were approached differently, as the structure remained in local care and completely related to its original place of construction [1].

Considering the advancement of geotechnical engineering, monumental structures started to be designed and executed. Among these structures, dams play an important role because the scope of their benefits and harms permeates large areas and populations. In this sense, the correct monitoring of this type of structure becomes fundamental.

The construction of the so-called “large” masonry dams, over 15 m high, had a great expansion during the 19th century, even before the appearance of concrete structures in the early 20th century. As an example of the use of concrete, we can cite a French engineering landmark, the Genissiat Dam, which is 106 m high and whose construction began in 1936 [1].

In order to produce a technical environment consistent with this type of mega-structure that was emerging, starting in 1928, industry players collaborated within the International Commission of Large Dams (ICOLD), in order to standardize construction, monitoring, and operational procedures. It was from this kind of joint effort that the rationalization of monitoring and the application of non-destructive techniques (NDT) were progressively refined [1].

Even though the knowledge gained about the condition of structures has enabled the development of regulations, codes, and standards, these documents lend themselves to aiding the supervision of the design and execution of the works, while offering little in terms of monitoring procedures for structures that are already in operation. Until recently, a more traditional but effective monitoring methodology was based on highly specialized professionals standing alongside the decision-makers to provide continuous feedback on the structure. This sometimes entailed the formation of a network of professionals who specialized in these issues, leading to the creation of structural monitoring reference offices [1].

This type of approach, dependent on reputable engineering firms, while effective, is quite financially onerous. It is a fact, however, that by making the structural monitoring process more accessible, civil engineering professionals will be able to analyze the structural stability of a given component in a faster, easier, and more economical way. It is precisely in this context that the present paper is inserted. By studying the use of acoustic sensor and artificial intelligence algorithms for monitoring dams, the goal is to provide automatic analyses that assess the state of these structures.

As shall be discussed in the following subsections, previous studies focused on different tools and algorithms to assess internal erosion. On the other hand, they did not cover an end-to-end methodological framework of the monitoring processing, covering the steps from sensor selection to data collection, data preprocessing, data processing, and structural scenario classification. Thus, the present paper will primarily focus on the study and development of a methodology for acoustically monitoring the stability of dams, considering the full monitoring workflow. Furthermore, a particular deep learning technique (autoencoders) will be applied, for the first time, to the specific task of acoustic monitoring of dams. Therefore, the main contribution of the paper is to provide a complete and easy-to-deploy monitoring solution for dams, mitigating the risks associated with this important type of structure. As indicated, the core of the methodology hereby presented are artificial intelligence algorithms.

In addition, the methodological framework proposed will be used to treat the dam monitoring data obtained in a scale model which was specifically built to test the algorithms. This dataset consists of acoustic data from geophones. Simply stated, geophones are sensors for sound waves transmitted through the dam body passively (environmental) or actively (synthetic hammers or pulses). The data from these sensors will be processed according to artificial intelligence algorithms, which will seek to identify patterns and anomalies, indicating possibles actions to be taken by the dam operator.

### 1.1. Dams and Their Pathologies

Dams are geotechnical structures mostly used for storing water for various purposes or, in some cases, for storing tailings from ore exploitation. The ICOLD estimates that embankment dams are the most common type of dam in operation, making up about 78% of all existing dams [2]. Among embankment dams, it is possible to distinguish soil or earth dams (about 83%) and rockfill dams (about 17%), either with a concrete face, clay core, or other type of waterproofing component [2].

Although well-constructed and well-maintained dams are safe structures, given the enormous damage associated with their rupture, these structures usually have a high aggregate risk. Thus, continuous monitoring of this type of structure is critical. Moreover, the availability of tools for a quick and technically correct decision making process is fundamental for the technicians responsible for such structures.

Normally, a large number of accidents in dams occur either during their first filling or during their first years of operation. However, there is also a significant number of accidents that occur later on [3]. Thus, in order to ensure an effective monitoring, it is interesting to know the potential failure modes of earth and rockfill dams.

Dams are considered highly complex engineering works, and it is reasonable to accept, at certain times, that the behavior verified in the field is not exactly the one predicted in the design [3]. According to Melo [4], the identification of potential failure modes (rupture) of an earth and/or rockfill dam converges on the general categories listed by Lafitte [5], namely:Hydraulic failures due to exceptional levels: includes, for example, overtopping and subsequent external erosion due to a spillway with insufficient discharge capacity, or even associated with gate damage or operating errors.Mass movements due to exceptional loads (except for the floods included in the previous item), inadequate material properties, or undetected geological singularities: includes, for example, slope instability (by limit equilibrium), deformations leading to overtopping, soil liquefaction, foundation instabilities, rapid subsidence associated with upstream slope sliding, and sliding of hillsides into the reservoir leading to overtopping.Internal erosion: includes, among others, development of piping in the dam core and erosion of foundation soils or joints (filling of discontinuities).

Regarding dam accidents, ICOLD has developed statistical studies in which it assesses the most frequent causes of rupture of this type of geotechnical structure [2]. Likewise, Foster et al. [6] analyzed the accident statistics associated with more than 11,000 dams. Although, due to lack of data, pre-1930 dams from Japan and China were not considered in that study, internal erosion can be regarded as the most frequent rupture cause. Internal erosion consists of the detachment and transport of soil particles by the action of forces caused by the flow of water through the dam body. Under these conditions, erosion occurs if the transport of the particles is geometrically possible through the voids and/or cracks existing in the earth or rock mass.

The study by Foster et al. [6] shows that about 50% of the failures occur due to internal erosion. It is clear, therefore, that among the most common failure mechanisms of earth and rockfill dams, this type of erosion is the one that generates the greatest concern because it develops internally and is, therefore, silent.

#### 1.1.1. Internal Erosion

Internal erosion can start in several ways, corresponding to different erosive mechanisms. For example, erosion can arise from concentrated leakage, that is, through preferential paths existing in the dam body or its foundation. These paths may be generated by stress transfer phenomena and consequent hydraulic fracturing or differential settlements, in the contact between the embankment and concrete structures or even with the foundation or deficient compaction, among others.

Another erosive mechanism, called regressive erosion, occurs in areas of free flow, where there are no filters, and corresponds to the increasing drag of soil particles in a progressive (sometimes even accelerated) and regressive manner, that is, the phenomenon starts downstream and moves upstream until establishing a connection. In this case, the phenomenon is called tubular erosion (or piping).

Suffusion is another possible mechanism and consists of the selective erosion of the finer particles of a soil, which are carried away through a coarser matrix of the soil. Soils with discontinuous particle sizes are more susceptible to this type of erosion as they may not be self-filtering. A fourth type of erosion can occur at the contact between two soils of very different characteristics. Contact erosion consists of the selective entrainment of fine particles from one soil through the coarser matrix of another soil. This erosion occurs when runoff is in the direction parallel to the interface between soils.

As can be seen, the initiation and development of internal erosion is composed of a series of complex phenomena and situations that are difficult to model and measure. In all the cases, it is noticed that the dam body loses homogeneity, and water starts to occupy the spaces once occupied by soil/earth or rockfill. In this context, artificial intelligence techniques can be used to simplify the process of analysis and verification, not of internal erosion, per se, but of the effects that this pathology causes in the response of the dam to external stimuli.

#### 1.1.2. Sensors Available to Monitor Internal Erosion

Most commonly, the standard external stimulus to which a dam is subjected is the upstream–downstream hydraulic potential difference. In this case, the dam’s response to this stimulus can be gauged by means of pore water pressure control. Thus, by means of piezometers, one can obtain an idea of how the inside of the dam is responding to the hydraulic load imposed on it. Note that this is only one of the possible ways of monitoring the internal erosion phenomena. What is sought with this method is the knowledge of the position of the water inside the dam body, given that water is the agent that fills the voids that are opening up in a piping situation.

Ferdos [7], for example, discusses the application of a tomography system based on gauging the electrical resistivity of the dam body. In summary, that author monitors the resistivity of several points in the porous medium and, by inverse simulation, relates the resistivity change to erosive phenomena. In a similar line of research, Ikard et al. [8] describe a test in which a saline pulse is percolated through the dam and the developed electrical potential (self potential) is measured as indicative of preferred flow paths.

Concerning a more general approach, Parekh [9] discusses the application of acoustic methods in combination with geophysical and electrical methods for detection and continuous monitoring of subsurface internal erosion initiation in its early stages. That author investigates the suitability of passive acoustic emission, self potential, and cross-hole tomography for suitability as long-term, remote, and continuous monitoring techniques for internal erosion and cracking of embankment dams.

Pueyo Anchuela et al. [10], on the other hand, studied the internal characterization of embankment dams using ground-penetrating radar (GPR) and thermographic analysis. Those authors indicate that GPR analyses at the embankments usually have the handicap of high clay content that precludes electromagnetic wave penetration. In the specific study case reported, however, they obtained resolution and extent of penetration using the different antennas in a sufficient manner, due to the absence of an inner waterproof unit [10]. This is not always the case, as the mineralogical characteristics of the core materials may be unknown, leading to inaccurate structural assessments of dams in general.

In addition, the growing concern about monitoring internal erosion can be demonstrated by a recent initiative by the United States Bureau of Reclamation (USBR) in collaboration with the United States Army Corps of Engineers (USACE). These major US agencies have launched a joint call for ideas for monitoring soil movement (erosion) within dams and foundations [11].

The prize competition titled “Detecting the Movement of Soils (Internal Erosion) Within Earthen Dams, Canals, Levees, and their Foundations” was the third Challenge Prize Competition conducted by the USBR, and awarded five proposals on the subject, which were repeated shear wave reflection imaging, unmanned aerial vehicles (UAVs) and underwater remotely operated vehicles (ROVs), permanent seismic monitoring system, repeated tracer testing/monitoring, and repeated/time-lapse superconducting quantum interference device (SQUID) magnetometer mapping.

From a practical point of view, in order to narrow the study of the present paper, non-destructive and do-no-harm solutions were considered. In particular, considering that acoustic monitoring has background both in the literature and in the industry, this was the selected acquisition sensor technology.

#### 1.1.3. Monitoring Dams’ Acoustic Data

While many studies have focused on the development of new sensors capable of identifying the phenomenon of internal erosion, more robust techniques of artificial intelligence that allow better exploration of the universe of existing data have received less attention.

Planès et al. [12] proposed a novel use of passive seismic interferometry to monitor temporal changes in earthen embankments caused by internal erosion. Those authors used the ambient seismic noise—i.e., ambient vibrations—propagating through the structure to assess its behavior. In order to check the applicability of their methodology, Planès et al. [12] monitored laboratory-scale and field-scale embankment failure experiments. Their main results revealed up to 20% reductions in surface wave velocity as internal erosion progressed in a canal embankment. In addition, the monitoring of a field embankment loaded to partial failure revealed a 30% reduction in averaged surface wave velocity. This way, they indicate that a time-lapse monitoring using their proposed methodology can be an efficient way to monitor internal erosion in dams.

According to Belcher et al. [13], the most common method of fault detection in a dam is visual inspection by a trained professional; however, this does not guarantee the detection of the fault at its early stage, which may not be enough to prevent collapse. Thus, in their study, these authors evaluated a way to monitor dams in real time with data analysis through machine learning. They indicate that it is possible to identify anomalies earlier and prevent dam collapse. In that study, the unsupervised machine learning technique called clustering was used.

The database of Belcher et al. [13] was obtained from “crack box” laboratory experiments, in which a dam is designed in a reduced scale and a fracture is promoted in the structure. The detection of the events was achieved by means of geophone sensors collecting data at 500 Hz. Time–frequency analysis using a non-overlapping fast Fourier transform showed that the events are separable (fault and normal) and that they provide unique signatures within the spectrum. Five sound features were selected to be analyzed for learning; these were zerocross, centroid, spread, RMS, and 85% rolloff. These features were normalized to have values between 0 and 1 in order to improve the efficiency of the clustering training.

The study evaluated clusters from 1 to 10 s and identified that three clusters obtained the best results [13]. The clustering algorithms evaluated were k-means clustering, hierarchical clustering, Gaussian mixture model, partitioning around medoids, and fuzzy c-means clustering. The performance of the five algorithms was evaluated using validation measures such as purity and silhouette width. The performance of the models had purity values above 0.823 and silhouette width values around 0.72, except for the Gaussian mixture model, which was much lower for both parameters (values close to 1 indicate better clusters for both parameters) [13].

The paper by Fisher et al. [14], similarly, is based on machine learning techniques and geophysical sensor technologies for identifying internal erosion and better understanding the structure of a dam. This investigation was performed from sensors on the surface of the dam.

Fisher et al. [14] adopt a workflow that is divided into four steps, data collection being the first one. The experiments used by those authors to represent internal erosion were fracturing and piping, both of which were acoustically monitored in the laboratory using geophones. The data from both experiments were filtered so that the dataset consisted mostly of normal data (80%) and a minority of anomalous data (20%).

The second step in [14] is preprocessing, in which noise elimination techniques were applied to the signals to better distinguish noise from anomalies (level 3 of the Haar wavelet transform was used). In addition, after denoising, the sound data were divided into several 1–10 s frames which would be processed by machine learning algorithms. In particular, nine sound characteristics were chosen as proxies for anomaly prediction (zerocross, centroid, spread, RMS, 85% rolloff, flatness, kurtosis, irregularity, and skewness). Before applying the machine learning techniques, data were normalized for better use. Still, along the second step, an algorithm for automatic selection of sound characteristics was used in order to keep the most relevant data for prediction (this selection was only carried out in the semi-supervised algorithm).

The third step in the methodology of Fisher et al. [14] was the application of machine learning algorithms, specially the support vector machine with the kernel radial basis function (RBF). Validation during training was carried out by the k-fold cross-validation technique. The fourth and final step is the presentation of the results, which are given as confusion matrices, tables, and graphs.

The study by Fisher et al. [14] found that both supervised and semi-supervised learning produce successful results with accuracy greater than 93%, and the difference in accuracy between supervised and semi-supervised is small. According to Fisher et al. [15], semi-supervised learning is important because one does not always have data on anomalies on a dam. Additionally, data can differ from what an anomaly would be at a specific dam.

Fisher et al. [15] study the internal erosion induced by fractures by using support vector machine in supervised learning and semi-supervised learning. The study in [15] also presented the computational efficiency for the mentioned workflow, which is the same as the one in [14] and was considered satisfactory. However, Fisher et al. [15] point out that for real-time data there should be further evaluation of the methods, because the amount of real-time data is much higher than what was obtained in the laboratory.

According to Fisher et al. [15], supervised analysis is a good first choice because it allows understanding the numerical behavior of normal and anomalous events. However, to perform this analysis, one needs to have evident data of abnormal and normal cases. This is not very accessible in real dams, but can be investigated through laboratory testing. Semi-supervised techniques, on the other hand, use only normal data, not requiring anomalous data, and are more feasible in ordinary monitoring. This technique tests the likelihood to identify what is out of the ordinary (or considered normal).

In their supervised learning approach, Fisher et al. [15] used all nine features (zerocross, centroid, spread, RMS, 85% rolloff, flatness, kurtosis, irregularity, and skewness) and an accuracy of 97% was obtained. In semi-supervised learning, an essential factor selection algorithm, ReliefF, was applied. This algorithm selected only two features that most influence the prediction. For the 3 s, 5 s, and 10 s frames, the zerocross and the RMS value were selected. With these two features, the predictions were made, and the accuracy of the model was 83%.

In the present section, the application of general machine learning techniques to the acoustic monitoring of dams has been discussed. On the other hand, in the artificial intelligence (AI) area, deep learning (DL) methods are part of the machine learning (ML) technique from deep neural networks (DNN). In short, DL consists of a set of algorithms that seek to shape high-level abstractions of data using several linear and nonlinear operations along multiple processing layers.

Among the promising DL techniques for anomaly detection, autoencoders show up. Zhou and Paffenroth [16] discuss the application of autoencoders that not only retain the ability to discover nonlinear features of a signal, but can also eliminate outliers and noise without access to any clean training data. Those authors present new approaches to anomaly detection, whose superior performance has been demonstrated on a selection of benchmark problems.

Regarding the application of autoencoders to dam monitoring, Chen et al. [17] used a time-series denoising autoencoder (TSDA) to represent the spatial and temporal features of the nodes by compressing high-dimensional monitoring data. Then, those authors used a network partitioning algorithm (NPA) based on spatial–temporal features obtained by the TSDA to monitor the dam itself. Therefore, autoencoders were not directly used to track the changes in the input signals, but to reduce its dimensionality and clean it.

More broadly, considering the health monitoring of buildings and bridges, Rastin et al. [18] proposed a new unsupervised deep learning-based method for structural damage detection based on convolutional autoencoders (CAEs). Their method aimed to identify and quantify structural damage using a CAE network that takes as inputs the raw vibration signals from the structure and is trained by the signals solely acquired from the healthy state of the structure. A similar approach, which relies on a regular deep autoencoder, will be considered in the present study.

In order to be able to use this DL approach and yet produce light and fast AI models, it is of interest to reduce the dimensionality of the input datasets. As pointed out by Jolliffe and Cadima [19], large datasets are increasingly widespread in many disciplines. In order to interpret such datasets, methods are required to drastically reduce their dimensionality in an interpretable way, such that most of the information in the data is preserved. This can be achieved, for example, by principal component analysis (PCA), which is is one of the oldest and most widely used techniques. Its idea is simple: reduce the dimensionality of a dataset, while preserving as much “variability” (i.e., statistical information) as possible [19].

In short, PCA finds an *n*-dimensional ellipsoid, also known as ellipsoid of residuals, which encompasses the points analyzed. Each axis of the ellipsoid represents a principal component, such that the smaller the axis, the smaller the variance will also be along that axis [20].The components are ordered such that the axis which explains most of the data’s variance is the first one, and the others follow a decreasing order of variance “explainability”.

Interestingly, autoencoeders with a single fully-connected hidden layer, a linear activation function, and a squared error cost function are closely related to PCA, the former being able to reproduce the results of the latter [21,22]. This way, autoencoders can also be used to decompose multidimensional datasets into smaller compressed elements, which then can be used for inference. It is worth noticing that PCA is limited to linear interactions, while autoencoders can learn complex nonlinear correlations between a dataset’s components.

Another interesting dimensionality-reduction technique is to obtain the Hjorth parameters of specific time windows in the input data stream [23]. Hjorth [23] defined three parameters for the characterization of electroencephalograms (EEGs), namely, activity (which measures the squared standard deviation of the amplitude, sometimes referred to as the variance or mean power), mobility (which measures the standard deviation of the slope with reference to the standard deviation of the amplitude), and complexity (which measures the excessive details with reference to the “softest” possible curve shape, the sine wave, this corresponding to unity).

From any dataset, the three Hjorth parameters (activity *a*, mobility mob, and complexity *c*) can be calculated as
(1)$$a=m_{0};\;mob=\sqrt{m_{2}/m_{0}};\;c=\frac{\sqrt{m_{4}/m_{2}}}{\sqrt{m_{2}/m_{0}}}$$
where
(2)$$m_{0}=\frac{1}{T}\int_{t-T}^{t}g^{2}\left(t\right)dt;\;m_{2}=\frac{1}{T}\int_{t-T}^{t}{\left(\frac{d}{dt}g\left(t\right)\right)}^{2}dt;\;m_{4}=\frac{1}{T}\int_{t-T}^{t}{\left(\frac{d^{2}}{dt^{2}}g\left(t\right)\right)}^{2}dt$$
and g(t) is the time series value at time *t* and *T* is the width of the observation window. Since we are dealing with discrete values, we can estimate the integrals above as the mean values of the squared signal g(t), the squared first difference of g(t), and the squared second difference of g(t), respectively. All the calculations are carried out within the observation window.

Hjorth parameters have been successfully applied in many acoustic monitoring scenarios [24,25], which make them good candidates to capture the most important characteristics of the input signal. In addition, the autoencoder DL approach demands a complementary tool to assign a threshold or to monitor the time-series of errors produced by the algorithm. Therefore, a CUSUM algorithm will be considered for the task and shall be described in the next subsection.

### 1.2. Time-Lapse Monitoring and Control Algorithms for Scenario Classifications

While performing a time-lapse monitoring of the dam, we need to distinguish between scenarios of structural status. Normally, a given metric is calculated sequentially and then changes in this metric are evaluated.

In general, it is of interest to detect change points that minimize the average detection delay while maintaining a given probability of false alarm [26]. The sequential change-point detection is widely applied in different fields including industrial quality control [27] and integrity monitoring in navigation satellite systems [28].

The context of the sequential detection can be characterized as follows: Let one consider a random phenomenon that can be observed sequentially in time. The behavior is supposed to be normal until a change that affects the behavior of the phenomenon occurs at an unknown time. Mathematically, let $\left\{x_{t}\right\},t=1,\ldots ,n=\left\{x_{1},\ldots ,x_{n}\right\}$ be a series of observations (data sequence) sequentially observed until time *n*, which is not fixed. In this case, $x_{n}$ is the last point added to the dataset [26].

It is also reasonable to suppose that during the normal state, the random variables $X_{i}s$ are identically distributed (not necessarily independent) according to the density distribution $f_{0}\left(.\right)$. This distribution changes whenever a change point occurs, and now can be reinterpreted as a postchange distribution $f_{1}\left(.\right)\neq f_{0}\left(.\right)$. Without loss of generality, it is possible to suppose that there exists only one change point at time *v*, such that $\left\{X_{1},\ldots ,X_{v-1}\right\}$ are identically distributed according to the prechange distribution $f_{0}$ and $\left\{X_{v},X_{v+1},\ldots ,X_{n}\right\}$ according to the postchange distribution $f_{1}$ [26].

Statistically, for all n≥1, the core of change-point detection algorithms is to sequentially test the null hypothesis, $H_{0,n}$, “no change point occurred before time *n*, that is v>n” against the alternative hypothesis $H_{1,n}$, “a change point occurred at the instant, that is, v≤n” [26].

In practice, however, the two distributions $f_{0}$ and $f_{1}$ are not always known. In this case, any approach based on likelihood is useless. Tartakovsky et al. [29] proposed a generalization for detection of change points in mean and/or variance by defining the score function $S_{t}$:(3)$$S_{t}\left(\delta ,q\right)=C_{1}Y_{t}+C_{2}Y_{t}^{2}-C_{3}$$
where $Y_{t}=\left(X_{t}-\mu_{0}\right)/\sigma_{0}$ is the centered and standardized data at time *t* under prechange regime, and
(4)$$C_{1}=\delta q^{2},\;C_{2}=\frac{1-q^{2}}{2},\;C_{3}=\frac{\delta^{2}q^{2}}{2}-\log\left(q\right),$$
with $\delta =\left(\mu_{1}-\mu_{0}\right)/\sigma_{0}$ and $q=\sigma_{0}/\sigma_{1}$ such that:δ is the mean difference between the postchange and prechange regime normalized by the prechange standard deviation. This parameter can be considered as the minimum level of change on the average that one desires to detect. Note that if the change point one seeks only concerns the variance $\mu_{1}=\mu_{0}$, then δ=0, and $C_{1}=0$.*q* is the ratio of variances between the prechange and postchange regime. This parameter can be considered as the minimum level of variance change one desires to detect. If the change point one seeks only concerns the mean (no detection on the variance), $\sigma_{1}=\sigma_{0}$, then q=1, so $C_{2}=0$.

This way, the based-score CUSUM statistic is defined recursively at time *t* as follows:(5)$$W_{t}\left(\delta ,q\right)=\max\left\{0,W_{t-1}\left(\delta ,q\right)+S_{t}\left(\delta ,q\right)\right\},\;t\geq 1;\;W_{0}\left(\delta ,q\right)=0.$$

The W-statistic presented in (Equation 5) can be defined according to the knowledge of the mean and variance in the prechange regime $\left(\mu_{0},\sigma_{0}^{2}\right)$ and of the detection objective (δ,q). It can also be noted that $W_{t}\left(\delta ,q\right)$ is adapted to the online (time-lapse) context because it is defined recursively. Notice also that in the prechange regime, the cumulative sum of $S_{t}$ up to time *t* is negative in expectation since $E_{0}\left(S_{t}\right)<0$; therefore, the statistic $W_{t}$ is often null. In the postchange regime, from the instant of the change point, $E_{1}\left(S_{t}\right)>0$ so that the cumulative sum of $S_{t}$ increases on average and begins to become positive [26].

Under postchange regime, knowing that the W-statistic tends to grow gradually, it is natural to reject $H_{0,t}$ when the statistic $W_{t}$ exceeds a threshold. This threshold has to be chosen according to an objective of false alarm rate that we denote by α [26].

Sahki et al. [26] argues that in the sequential detection context, the conventional threshold used in the literature is constant over time. This constant threshold h(α) is determined such that the risk to detect an alarm at time *t* is controlled by α:(6)$$P_{0}\left[W_{t}\left(\delta ,q\right)\geq h\left(\alpha \right)\right]\leq \alpha ,\;t\geq 1$$

The classical method [26] used to set a constant threshold is based on the following Wald inequality [30]: α≤exp(−h). Therefore, the Wald constant threshold $h^{W}\left(\alpha \right)$ is given after fixing the tolerated α, by respecting
(7)$$h^{W}\left(\alpha \right)=-\ln\alpha$$

Literature [31,32] indicate that α can be interpreted as the inverse of the mean time between false alarm (MTBFA), which evaluates the average number of observations before triggering a false alarm. In this case, the higher the MTBFA, the lower the real risk of false alarm (lower is α). This way, selecting smaller values of alpha increases the sensibility of the algorithm.

Other threshold values can be chosen, such as the empirical constant threshold, conditional empirical instantaneous threshold, empirical instantaneous threshold, and the dynamical empirical instantaneous threshold [26]. On the other hand, for the present paper, the simple Wald constant threshold is considered.

If we are monitoring a single metric of the acoustic signal, we can rely on the alarm provided by the CUSUM algorithm. On the other hand, when we have multiple metrics being monitored, we can use fuzzy logic to enhance the alarm triggering and scenario classification.

Regarding fuzzy logic, the probability theory is an efficient tool to deal with uncertainties, as long as it is applied to situations whose characteristics are due to random processes. On the other hand, for cases where the uncertainties are due to imprecision, the theory of fuzzy sets presents itself as an excellent mathematical tool [33]. In fuzzy logic, features are no longer binary (Boolerian logic) but have a degree of association. In other words, the fuzzy set is a continuous interval of type [0, 1] while the Boolerian set is of type 0, 1. In the broad area of geosciences, fuzzy logic has been widely used in research in the areas of geotechnical engineering, surface and subsurface hydrology, and hydrocarbon exploration [34]. In addition, Demicco [34] noted its application in areas such as groundwater risk assessment, seismology, landform formation, and sediment deposition.

In the case of scenario classification, we can use fuzzy logic to model the alarm triggering of CUSUM algorithms previously described. This way, it is possible to build a smooth scenario classification procedure.

It is worth noticing that a first idea would be to explore the application of fuzzy logic algorithms by themselves, as the core engines of the monitoring framework. On the other hand, preliminary studies carried out by the authors revealed that this approach was not able to fully capture the nonlinearities and particularities of the input acoustic signals considered in this study. More broadly, the core idea in the methodological framework hereby proposed is to consider a tool capable of detecting even slight changes in the input acoustic signal, despite all the possible environmental interferences.

In this context of time-series modeling, deep neural autoencoders are among the gold standard alternatives presented in the literature. This type of AI technique is capable of learning not only how each sensor data stream behaves, but also the correlation between them. This is exactly what is needed in this internal erosion monitoring problem. In the next section, the methodological framework hereby developed is presented.

## 2. Methodology

This section presents the steps necessary to analyze acoustic data from the monitoring of dams, ultimately providing its structural status. A diagram is presented in Figure 1 to illustrate these methodological steps.

In the case of the present paper, data collected from a scale model are analyzed by semi-supervised techniques, which allows the monitoring of the evolution of acoustic signals measured in the dam.

This choice is due to the fact that a model calibrated in a supervised manner could be unable to be generalized to a situation of real dam monitoring. Unlike in other fields, such as image recognition, where learning transfer is possible, in the case of acoustic signals detected at dams, each structure has a potentially distinct characteristic behavior. This makes semi-supervised approaches, which use the local data itself for their calibration, to be more scalable and generalizable. Thus, the semi-supervised deep autoencoder technique will be the primary algorithm to be used.

The data flow procedure considered in the present paper coincides with the methodological steps presented in Figure 1. In short, the calibration flow is as follows:Collect data from low-cost Raspberry Shake seismic stations during “normal” structural conditions (may be the current situation).Preprocess the data collected in order to reduce its dimensionality, i.e., calculate the Hjorth’s parameters for each data component.Split the “normal” dataset into train and test, to evaluate if the algorithm is able to indicate that the data is, in fact, “normal”.Process the Hjorth’s parameters for each training data component by calibrating an autoencoder. This DL algorithm should be able to reproduce the structure of the input data.Perform a scenario classification on the test data, checking the stability of the predictions, which should indicate a “normal” state of the dam. This scenario classification step involves passing the time series of errors outputted by the autoencoders though a CUSUM algorithm and a fuzzy logic classification code.

If these steps are successfully completed, one shall proceed to the monitoring flow, which can be summarized as follows:Collect data from low-cost Raspberry Shake seismic stations during operation conditions.Preprocess the data collected in order to reduce its dimensionality, i.e., calculate the Hjorth’s parameters for each data component.Process the Hjorth’s parameters for each training data component by using the calibrated autoencoder to reproduce the input dataset.Perform a scenario classification on the data just collected, which should indicate the state of the dam. This scenario classification step involves passing the time series of errors outputted by the autoencoders though a CUSUM algorithm and a fuzzy logic classification code.

Each of the steps indicated will be explored in the next subsections. The monitoring flow will be tested by applying it to a study case, also described in the present section.

### 2.1. Preprocessing

In this step, dimensionality reduction techniques were applied to decrease the number of components to be modeled. This process starts from the idea that an acoustic signal, for example, can be represented by specific metrics that capture its behavior. Thus, for a sample taken at *x* Hz, instead of dealing directly with the *x* data collected per second, one can apply a transformation that provides metrics, which will be modeled. The data collected via the Raspberry Shake sensors, which will be further characterized, for the scale model scenarios had a sampling frequency of 100 Hz which, according to Nyqist’s theorem, allows us to study signals up to 50 Hz frequency well. This is sufficient for our applications, whose noise source is passive. Hjorth’s parameters [23] were chosen as representative features for each second of data collected (i.e., 100 samples).

### 2.2. Processing

In this step, the monitoring of the evolution of the representative metrics of the acoustic signals was performed. Assuming that during the beginning of the monitoring the structure is far from rupture, the metrics have a specific behavior called “normal”. After the time-lapse monitoring, the metrics start having a different behavior from the one initially observed, indicating that changes are occurring in the dam body.

To model normal behavior, the deep learning technique called autoencoder was used. This type of entity seeks to understand the relationships between the various input components of the algorithm in order to be able to reproduce them. Thus, by calibrating the autoencoder with so-called “normal” data, the error of the algorithm, defined as the average of the absolute value of the difference between the input and output data, tends to be small for this data. When anomalous data is entered as input, the autoencoder fails to predict the outputs consistently, which is a direct result of the differences between the structure of the new data and the “normal” data.

During our modeling we used signals from multiple sensors, which allows the algorithm to learn not only the individual evolution of each component, but also the correlations that exist between them.

### 2.3. Scenario Classification

From monitoring the error of the autoencoders calibrated in the previous step, state scenarios of the structure can be defined. We chose to delimit four scenarios, namely, Green, Yellow, Orange, and Red.

As indicated, a CUSUM technique is used to track the cumulative sum of a statistic to check for changes in the series of reconstruction errors from the algorithm [29]. The threshold for the cumulative statistic is chosen as Wald’s constant threshold, i.e., equal to −lnα.

As indicated [26], α is the inverse of the mean time between false alarms. The value of α needs to be chosen in a way that the MTBFA can be large, which corresponds to a big probability of detecting a correct alarm (1 − α). On the other hand, if the system is too sensitive, alarms will be triggered unnecessarily. It is also important to account for short-time interferences, which could trigger the system. In order to do so, it is possible to increase α. In the present paper, each monitoring step in the algorithm considers a window of 1 s of acquired data. Thus, 100 s was chosen as the mean time between false alarms, leading to α=0.01. This value could be changed to adjust the sensitivity of the detection, reaching an optimal value whenever most of the alarms triggered come from important change events in the data stream.

As for the types of scenarios, the Green scenario assumes that the behavior is normal and there is no need for interventions. The Yellow behavior, on the other hand, indicates that important changes have already been initiated, so that an analysis should be performed in order to seek the best solutions for intervention in the structure. When the Orange behavior is reached, one should act immediately with urgent measures in order to stop the degradation of the structure. Finally, when the scenario is Red, one must clear the areas affected by the structure, because it is facing an imminent rupture scenario. Fuzzy logic techniques will be used to perform this classification.

### 2.4. Monitoring a Scale Model —SM

In this subsection, the scale model built to validate the methodological framework hereby developed is described.

#### 2.4.1. Experimental Setup—Model Assembly

The experimental setup was built in the GeoFluxo Laboratory, located in the underground floor SG12 building of the University of Brasilia, Campus Darcy Ribeiro (latitude −15.7649°, longitude −47.8727° and elevation 1045 m). The construction system of the SM apparatus involves the assembly and/or construction of some components, such as the concrete block (base for fixing the model), the SM simulation aquarium, the wetting system, the water collection system, and the monitoring system.

The concrete block was the first component built, which served as the base to support the fixation of the aquarium for the SM simulation. It was built directly on the laboratory floor with dimensions of 1.26 m in length, 0.50 m in height, and 0.60 m in width, corresponding to a structure of about 0.38 m^3^ of concrete.

The wetting system was conceived to guarantee a continuous supply of water during the simulation of the filling of the upstream reservoir of the dam. This system consists of a 290 L reservoir with an outlet of four 25 mm valves arranged in parallel, which were coupled to 5/16” × 1.5 mm hoses with reduction to 1/4” × 1.0 mm. This composition was made so that the filling of the reservoir in the aquarium would occur homogeneously and in a way that no waves would be created on the upstream face of the dam.

The water catchment system was designed to reduce the noise caused by the water exiting the model and to allow the researchers to reuse the water that passes through the SM. This system is composed of three parts: a plastic gutter that conducts the water passing through the SM, a reservoir with a filter in the lid for monitoring the possible carrying of materials during the experiment, and a hose that is used to empty this reservoir when necessary.

The SM dam, on the other hand, was assembled inside the simulation tank. For the assembly of each dam, 20 kg of a sandy soil with a bimodal granulometry was employed. In short, a total of 16 kg of coarser material (granulometric range corresponding to 0.85 mm–4.75 mm) was mixed to 4 kg of a finer material (particle sizes ranging of 0.25 mm–0.60 mm). To this mixture, 980 mL of water was added to enable the molding of the dam at a gravimetric humidity of 5%.

The model consists of five material layers, each of which had a thickness of 0.03 m. Since the dam body has a trapezoidal cross-section, each layer has a different weight. From the bottom to the top, each layer had a mass of 6.8 kg, 5.2 kg, 3.9 kg, 2.6 kg, and 1.3 kg. It is worth mentioning that all the layers were compacted manually to obtain the desired shape and the specific mass of 1.7 g/cm^3^. The geometry of the dam can be seen in Figure 2.

After completion of the five layers, a support was placed on the dam crest to ensure the coupling of the seismological station to the dam body. The support in question was made using 3D printing and the material used was polylactic acid (PLA) filament, as seen in Figure 3.

#### 2.4.2. Seismic Monitoring System

The seismic monitoring system is composed of three seismological stations. Each of the stations encompass the solution provided by Raspberry Shake, which offers a combination of sensors, hardware, and software in a single device. This solution is composed of multi-component (three orthogonal) sensors formed solely by geophones (Raspberry Shake 3D), plus a digitizer system and real-time communication.

The cost–benefit of this approach is interesting for low-cost and high-fidelity systems such as the one considered in the present study. In addition, literature reveals that the Raspberry Shake has been successfully applied to a number of problems, such as assessing rockfall activity in the Alps [35], debris flow identification [36], and earthquake monitoring [37], among many others.

The stations were positioned ensuring parallelism between the north of the stations with the main direction of the SM water flow (i.e., from upstream to downstream), which implied an angle of 102° to geographic north.

The three stations were arranged in a line, as shown in Figure 4, with one station positioned at the crest of the SM (AM.R7D9F), the second on the concrete block (AM.RFBA7) next to the SM, and the third positioned directly on the floor of the laboratory (AM.R016A).

It should be noted that a number of precautions were observed to ensure a good coupling of the seismological stations to the underlying surfaces during the assembly of the acquisition system; these were the correct positioning of the sensors considering the central position of the bubble level of the stations, and the fixing of the stations to the underlying surfaces with the aid of screws aiming at the largest area of contact of the sensor with the underlying layer. After positioning the sensors, the stations could be connected to the local electrical and internet networks for data acquisition.

Two different testing dams were built: a homogeneous one and another with an anomaly. To simulate the existence of an anomaly in the dam body, a PVC tube of 0.015 m internal diameter and 0.524 m in length was placed at a height of 0.090 m (from the bottom of the dam). This discontinuity can allow a water flow inside the dam from upstream to downstream.

Since it is desired to model the appearance of an anomaly during dam operation at a given reservoir quota, the upstream end of the PVC tube was sealed with a latex material to prevent water flowing into the pipe prior to the desired moment.

To assist in the analysis of the acquisition data, during the monitoring of the dam, several other pieces of information were collected, such as the identification of some acoustic unforeseen noise sources and the monitoring of the height of the reservoir. This information is crucial to define the training and testing phases of the methodological framework hereby developed.

The tests consisted basically of three stages, which were filling the upstream reservoir, maintaining a given quota, defined as the service level, and, finally, lowering the upstream reservoir.

In short, the filling stage started when the four valves were opened. The first water jets were discarded to avoid the formation of bubbles inside the hoses, which could prevent a constant flow of water. Once the continuous water flow was guaranteed, the inlet flow rate to the upstream reservoir was regulated. The low flow rate was crucial to ensure a laminar flow and to avoid the formation of waves that could erode the upstream face of the dam. The laminar flow was kept constant until the service level was reached.

When service level was reached, if necessary, one or more valves could be opened and closed to regulate the outflow of the water from the reservoir, allowing the maintenance of the service level. The final stage was the lowering of the water level, so all the valves were closed and the researchers waited for the complete emptying of the upstream reservoir.

## 3. Application of the Methodology to Monitor the Scale Model

For the cases under analysis, three Raspberry Shake sensors were available, namely,

AM.R7D9F on the crest of the reduced model;AM.RFBA7 on the concrete block;AM.R016A on the GeoFluxo laboratory floor.

Each of these sensors is capable of acquiring three components of the acoustic signal. Due to a reading problem, the vertical component was not recorded for sensor AM.R7D9F, so that, in total, there were eight signal components for all sensors and directions available.

From the experiment log, the times at which each type of data was collected could be defined. In particular, it is possible to separated the intervals when

The reservoir upstream of the dam was filling;The reservoir upstream of the dam was at a fixed top quota;The reservoir upstream of the dam was emptying.

It was necessary to build two dam models, one homogeneous and another one with a PVC tube inside it. This way, by combining the time intervals above with the two dam configurations, there are three conditions of dam integrity/flow:Homogeneous dam (homogeneous model and upstream reservoir at fixed top quota);Dam with heterogeneity and no flow through the pipe (model with sealed PVC tube and upstream reservoir at fixed top quota);Dam with heterogeneity and flow through the pipe (model after unsealing the PVC tube and upstream reservoir at decreasing top quota).

From Figure 5, Figure 6 and Figure 7, the acoustic data collected using the Raspberry Shake sensors are presented. One may notice that each plot has the correspondent component indicated on its right-hand side (EHE, EHN, and EHZ, the latter being the vertical component and the other two, orthogonal to EHZ).

For all eight components, the Hjorth parameters were calculated. It should also be noted that the data was scaled according to the StandardScaler() function from the sklearn package. This allows the scaled data to have zero mean and unit standard deviation.

For each of the eight components of acoustic data, we have three Hjorth parameters. This way, visualizing this data is not straightforward, as an eight-dimensional plot would be needed to check how the values of each Hjorth parameter vary for each sensor when the dam structural status changes. We can use principal component analysis (PCA) [20] to reduce the dimensionality of this eight-fold dataset, allowing it to be plotted. It is worth noticing, on the other hand, that the full eight-fold scenario is used in the classification algorithm.

Therefore, Figure 8, Figure 9 and Figure 10 present the first and second principal components of the eight-dimensional dataset for each Hjorth parameter, obtained by using sklearn’s implementation named “PCA”. The datasets were also separated according to the experimental arrangement of the dam. To make the visualization of the spreading of the points easier, a convex hull (the smallest convex set that contains the points [38]) was drawn using scipy’s implementation of the “ConvexHull” class.

Just by analyzing Figure 8, Figure 9 and Figure 10, it can be seen that the behavior of the principal components for each Hjorth parameter changes according to the structural status of the dam. For example, it is easy to see in Figure 8 that both the principal components for the “homogeneous” and “heterogeneous without water flow” configurations are similar, while a major change is observed for the “heterogeneous with water flow” scenario. In the latter case, the points tend to align according to an inclined axis, differently from the other cases, where the data are distributed around a circular area. Similar analyses can be carried out for Figure 9 and Figure 10.

Despite this initial assessment of the changes of the Hjorth parameters according to the dam configurations, the main methodological strategy of this study is to use autoencoders to quantify these changes. Thus, at this point, the separation into training and test data is complete. Importantly, to verify the adequacy of the data collected and the algorithms, 15% of the data considered normal was added to the test data. This helps in measuring the detection of false positives.

Since we are dealing with an autoencoder algorithm, we use the data corresponding to the homogeneous case for calibration.

In this step, the autoencoder was assembled using the Keras and TensorFlow packages. The selected autoencoder is a deep neural network which has three internal dense layers; the first with ten nodes, the second with two, and the third with ten nodes. A last output dense layer was added to retrieve the shape of the input vector. This architecture is in accordance with the assumptions for an autoencoder.

For each Hjorth parameter, a deep learning network is calibrated considering a 5% validation split. An exponential linear unit (“elu”) activation function was chosen and the kernel initializers were all set to “glorot_uniform”. The model was trained for a maximum of 100 epochs with batches of size 10. The loss function chosen was the mean squared error (“mse”) and the Adam optimizer algorithm was selected (“adam”).

The functions that are needed to check the series of reconstruction errors for each Hjorth parameter were implemented. The point metric proposed by Tartakovsky et al. [29] and presented in Equation (Equation 3) was implemented in Python, which allows to evaluate changes in the mean and standard deviation of values in a series. In addition, the $W_{t}$ statistic presented in Equation (Equation 5) was implemented as well, allowing a sequential application as new data arrive.

In the present paper, the CUSUM algorithm and score metric $S_{t}$ look for changes of the order of 2.5% in the mean only, disregarding changes in the standard deviation of the series’ values. Therefore, we set $\mu_{1}=1.025\mu_{0}$ and $\sigma_{1}=\sigma_{0}$ to use Equation (Equation 5).

In order to classify the scenarios, as indicated, the CUSUM statistic was calculated and should be compared to a threshold. If the statistic is greater than this threshold, then it is understood that there has been a change in the series. As indicated, this threshold value will be taken as −ln(0.01)=4.6052, according to Wald’s inequality [26].

For the purposes of this research, from the point of view of fuzzy logic, we can consider that the error indicators for each Hjorth’s parameters will be classified as of the types “low” and “high”. The change between one and the other will be given by the limit obtained by Wald’s inequality. The output scenarios will be of the types “Green”, “Yellow”, “Orange”, and “Red”.

The fuzzy environment was built with the skfuzzy python package. In addition, the fuzzy sets were built considering the membership functions as sigmoids. It is now possible to define the fuzzy rules that will direct the classification. In summary, the rules are as follows:If none of the indicators are of type “high” (i.e., greater than the threshold imposed by Wald’s inequality), the system state is green;If only one of the indicators is of the “high” type (i.e., greater than the limit imposed by Wald’s inequality), the state of the system is yellow;If only two of the indicators are of the “high” type (i.e., greater than the limit imposed by Wald’s inequality), the state of the system is orange;If all indicators are of the “high” type (i.e., greater than the Wald inequality threshold), the state of the system is red.

Thus, it is possible to plot the classified values over time. For simplicity, in Figure 11 is plotted the evolution of the error metric for mobility and the color of the dot will reflect the class it belongs to.

It can be seen from Figure 11 that the algorithm was able to detect, in a completely semi-supervised and highly effective way, the changes in the dam of the scale model. Note that the plotted vertical lines indicate three scenarios; from left to right:Period used for benchmarking the algorithm (normal);Period used for benchmarking the algorithm (with heterogeneity and no flow);Period used for measuring the algorithm (with heterogeneity and with flow).

The algorithm was able to quickly detect the changes between dam integrity scenarios, which shows the suitability of the methodological proposal. It is also important to study the contribution of each of the reconstruction error series to the final result of the system state classification. Thus, consider Figure 12, which plots the metrics for the error evolution of each Hjorth parameter as well as the Wald’s constant threshold.

It can be seen that the evolution of errors for the parameters mobility and complexity are linked to the level of homogeneity of the dam, while for the parameter activity the highest correlation is with the existence, or not, of water flow. The flow tends to generate higher sound amplitudes, because the sound of the water moving tends to become louder. This is exactly what the reconstruction error in the activity parameter indicates, because it is a measure of the energy of the signal. This also reinforces the principal component analysis discussion carried out previously.

Even though the methodological framework proposed (and its algorithms) showed good agreement with observed phenomena, some issues may still show up during a field deploy of the system. For example, an important issue which may show up is the unavailability of one or more sensors. Even after pointing out to the dam operator that there exists broken data links, the system must be able to continue outputting its predictions. A possible workaround is to train multiple autoencoders, one for each combination of possibly available input parameters, and activate the corresponding model according to the inputs received. For an eight-fold input data stream, a total of 28−1=∑n=188n=255 autoencoders would need to be trained and evaluated. The autoencoders which were not able to predict accurate alarms using the scale-model results should be discarded and, in these cases, besides indicating which data links are offline, the system would stop outputting predictions.

Another issue which needs to be assessed as soon as the “normal” status datasets are recorded is whether or not the autoencoder architecture hereby proposed suits the problem. The 10–2–10 three-layer model may be insufficient to accurately model the “normal” datasets, requiring the code to be updated. In this case, by splitting the “normal” datasets into train and test subsets and using k-fold cross-validation, it is possible to look for a better model.

Finally, fine-tuning the Wald’s constant threshold, which depends on α, can also enhance the capabilities of the methodological framework. By increasing (decreasing) α, the algorithms become less (more) susceptible to triggering alarms, which may be of interest when the associated rupture consequences of the dam are highly critical.

## 4. Conclusions

Although well-constructed and well-maintained dams are safe structures, given the enormous damage associated with their rupture, they are usually structures with a high aggregate risk. Thus, continuous monitoring of this type of structure is critical. For that, artificial intelligence methods, especially deep neural autoencoders, have been shown to be effective tools.

In the present paper, the acoustic data collected by geophones installed on a scale model dam were modeled to assess its structural status. The acoustic dataset was preprocessed to reduce its dimensionality by obtaining the Hjorth parameters for each second of acquired data.

Each Hjorth parameter obtained for every sensor was used to calibrate an autoenconder. Then, the reconstruction error of each autoencoder was used to monitor how far from the original (normal) state the acoustic signature of the scale dam was. The time series of reconstruction errors were combined with a cumulative sum (CUSUM) algorithm, which indicated changes in the sequential data collected. To combine the output of the CUSUM algorithm for the three autoencoders, a fuzzy logic framework was used to predict the status of the structure.

The methodological framework hereby developed indicated that the evolution of errors for the parameters mobility and complexity are linked to the level of homogeneity of the dam, while the parameter activity had a highest correlation with the existence, or not, of water flow. This way, the algorithm was able to predict, in a semi-supervised way, all the changes in the dam.

## Figures and Tables

**Figure 1 sensors-22-02482-f001:**
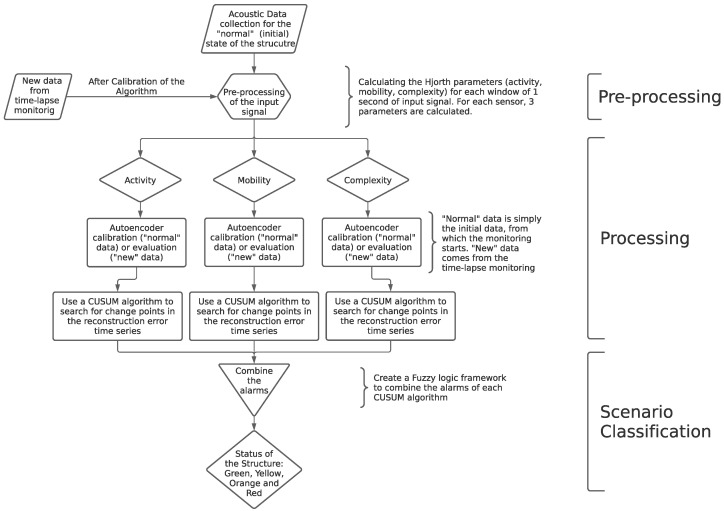
Methodological framework.

**Figure 2 sensors-22-02482-f002:**
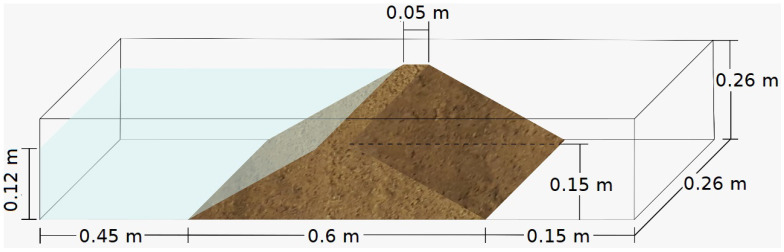
Geometry of the SM dam.

**Figure 3 sensors-22-02482-f003:**
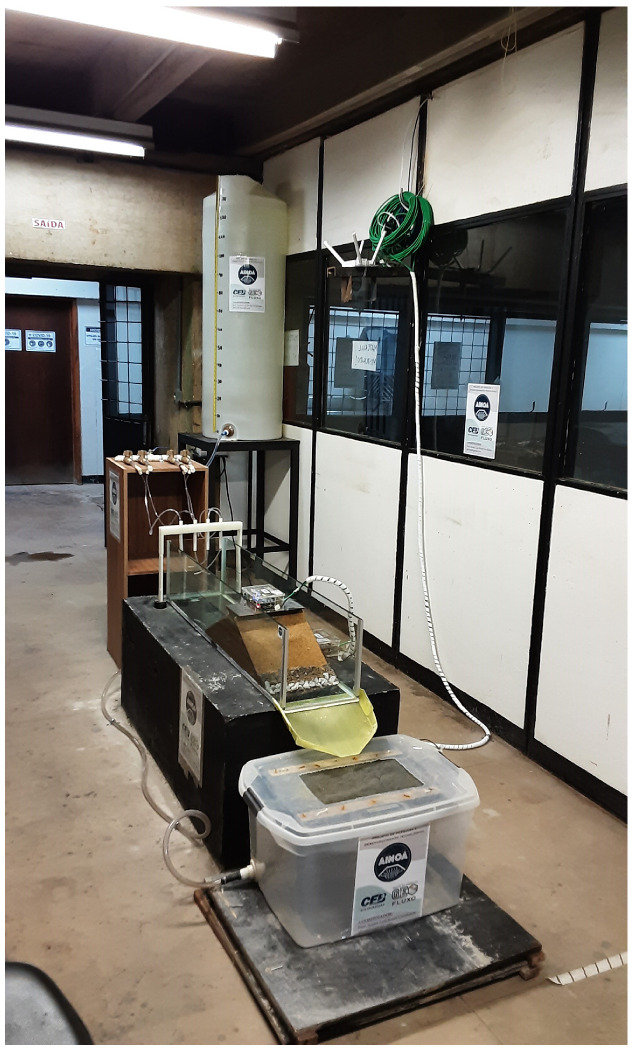
Lateral view of experimental setup.

**Figure 4 sensors-22-02482-f004:**
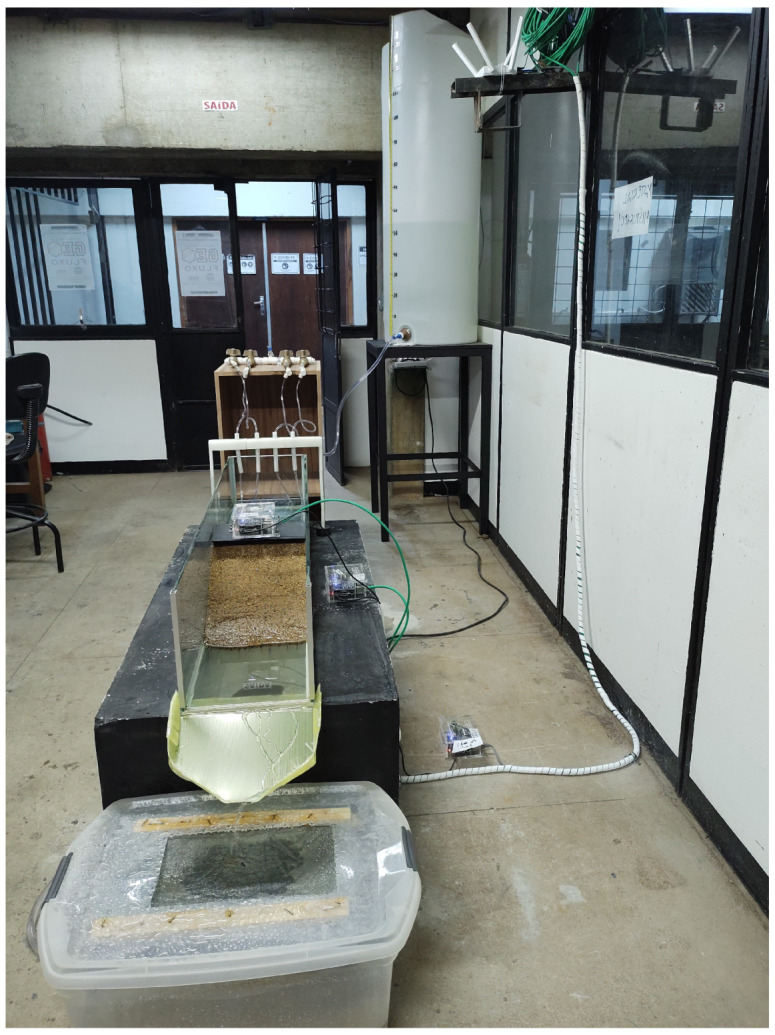
Frontal view of experimental setup.

**Figure 5 sensors-22-02482-f005:**
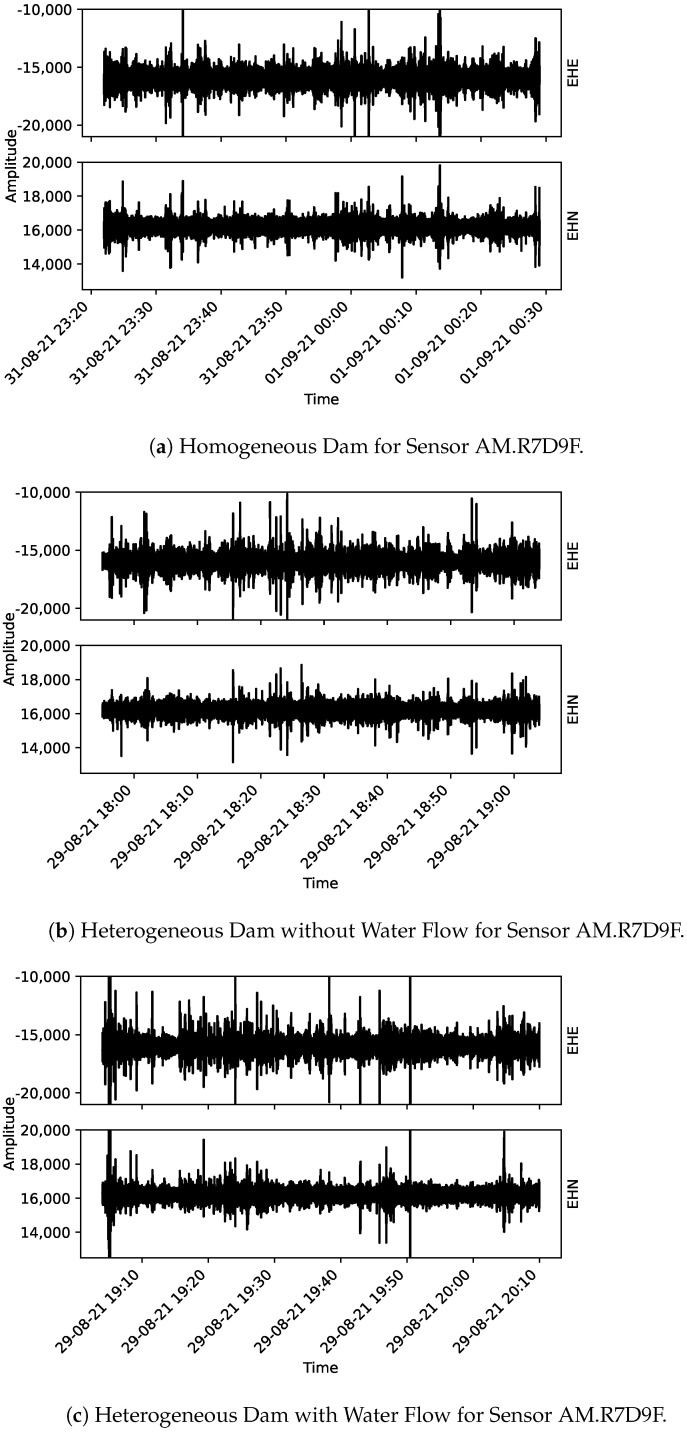
Collected data for sensor AM.R7D9F. From (**a**) to (**c**) the three monitoring stages are presented: homogeneous dam, heterogeneous dam without flow and heterogeneous dam with flow, respectively.

**Figure 6 sensors-22-02482-f006:**
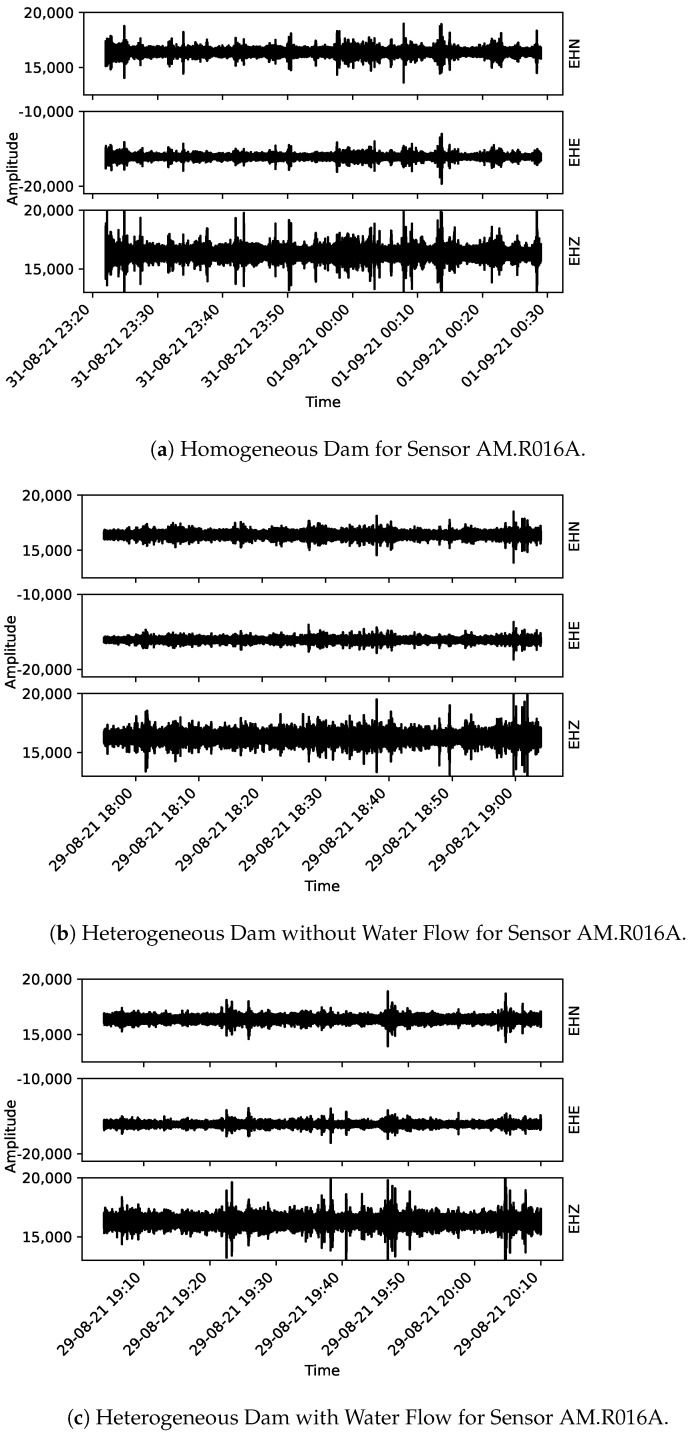
Collected data for sensor AM.R016A. From (**a**) to (**c**) the three monitoring stages are presented: homogeneous dam, heterogeneous dam without flow and heterogeneous dam with flow, respectively.

**Figure 7 sensors-22-02482-f007:**
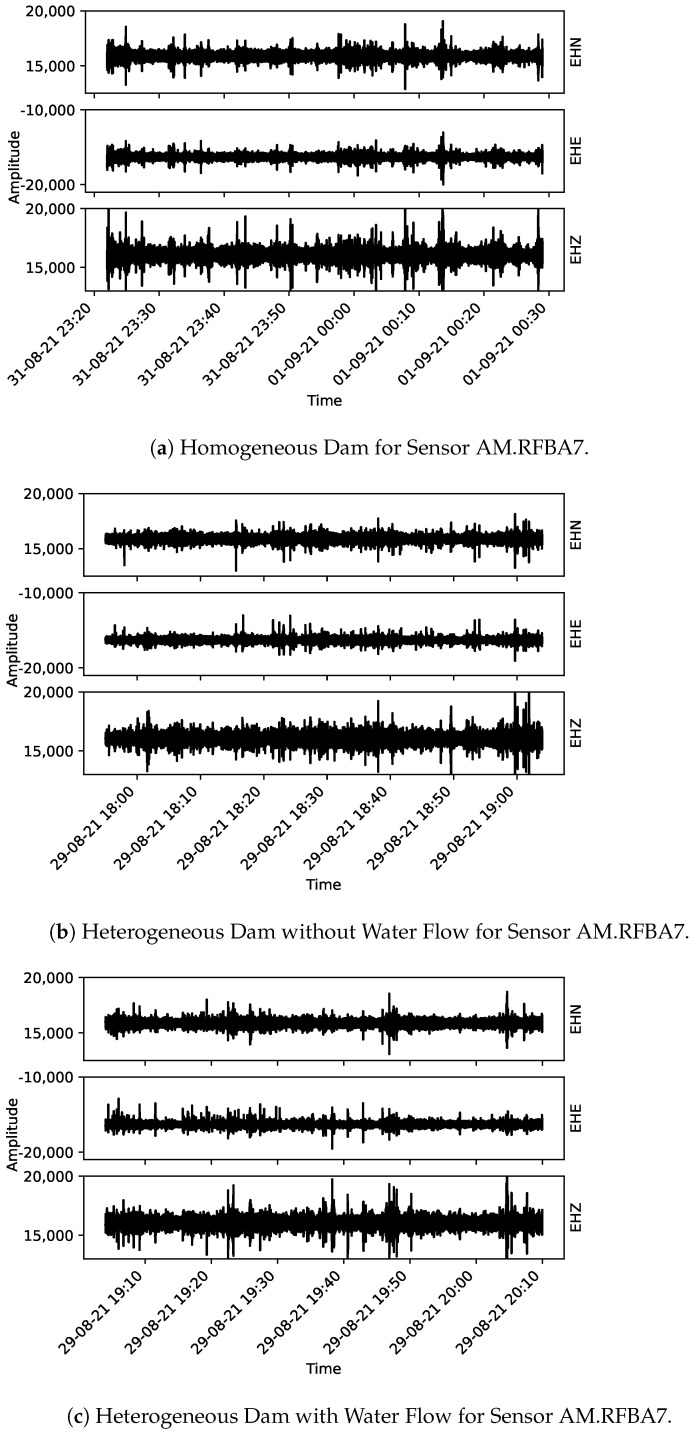
Collected data for sensor AM.RFBA7. From (**a**–**c**) the three monitoring stages are presented: homogeneous dam, heterogeneous dam without flow and heterogeneous dam with flow, respectively.

**Figure 8 sensors-22-02482-f008:**
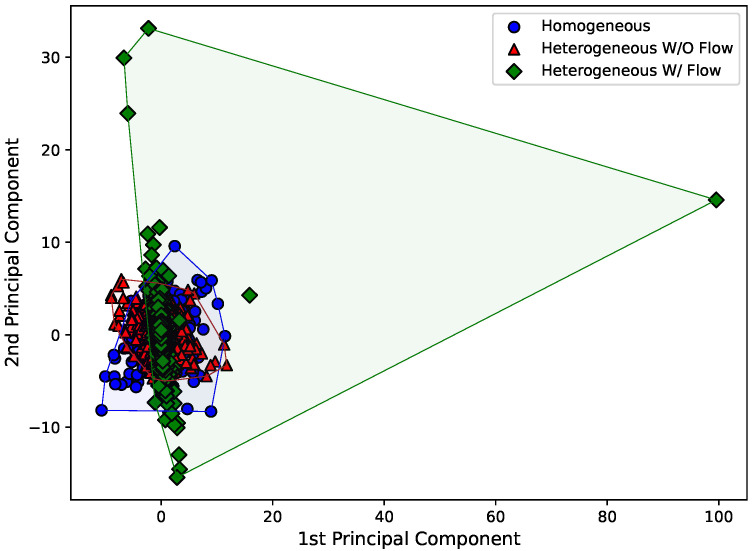
First and second principal components for the activity Hjorth parameter considering all the eight sensors.

**Figure 9 sensors-22-02482-f009:**
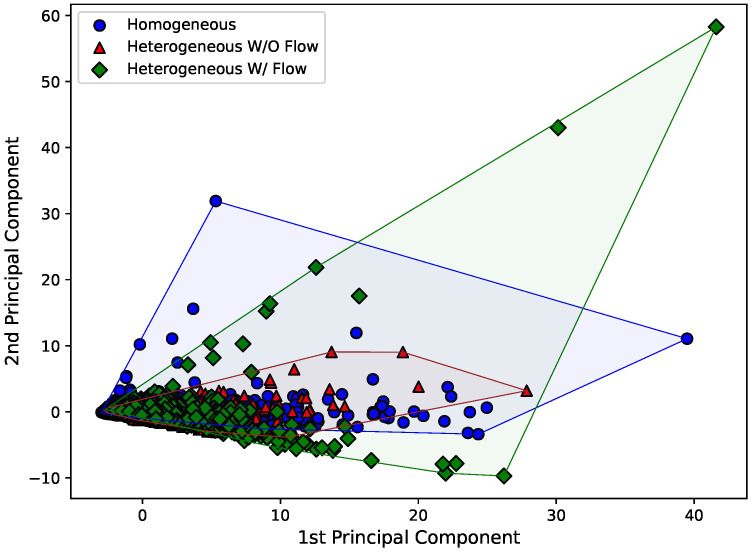
First and second principal components for the mobility Hjorth parameter considering all the eight sensors.

**Figure 10 sensors-22-02482-f010:**
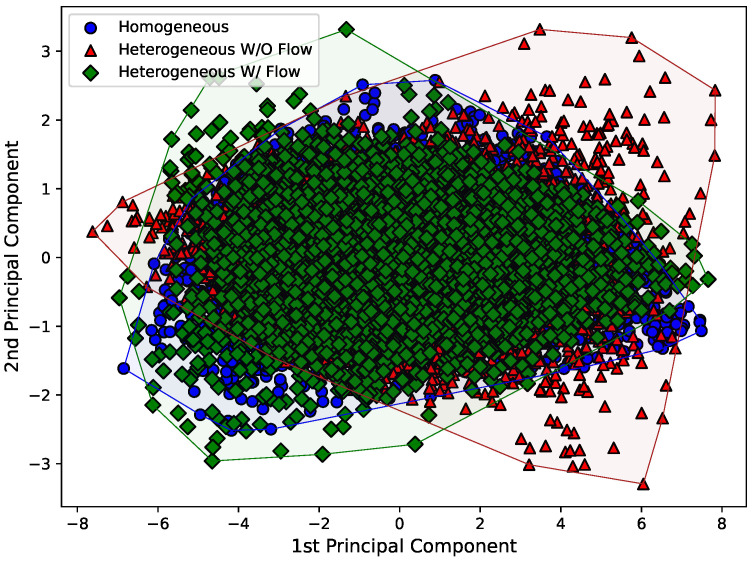
First and second principal components for the complexity Hjorth parameter considering all the eight sensors.

**Figure 11 sensors-22-02482-f011:**
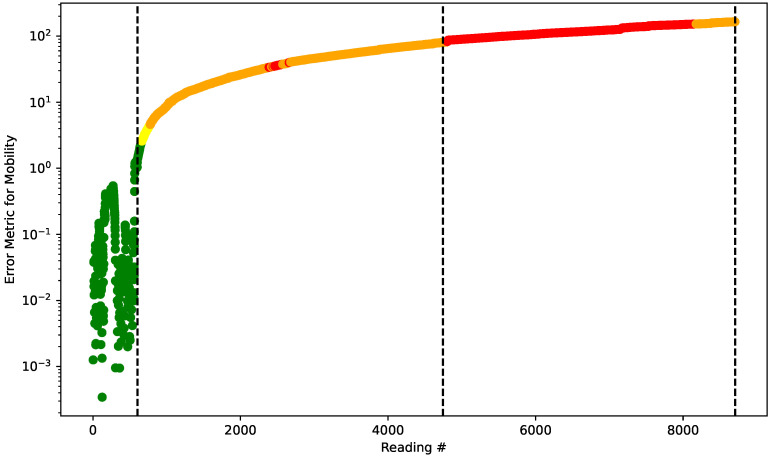
Evolution of the error metric for mobility and the correspondent status of the system.

**Figure 12 sensors-22-02482-f012:**
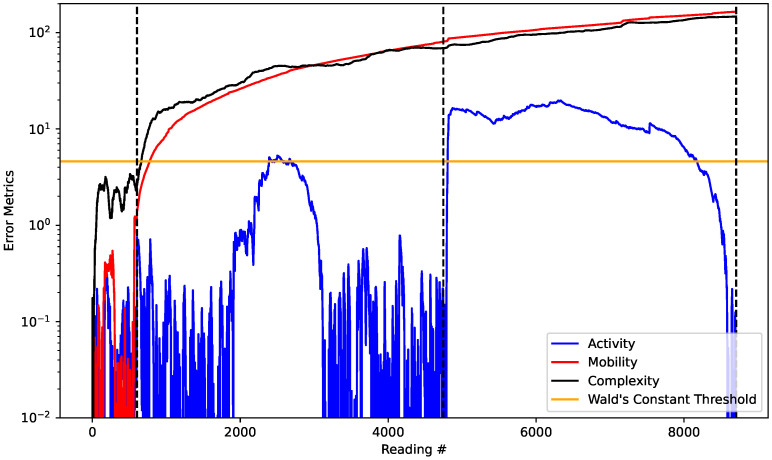
Evolution of all the error metrics of the Hjorth’s parameters.

## Data Availability

The data used are available upon request to the corresponding author.

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
