# Peer review of "Structural Health Monitoring of Dams Based on Acoustic Monitoring, Deep Neural Networks, Fuzzy Logic and a CUSUM Control Algorithm"

_sensors, 2022, doi:10.3390/s22072482_

Round 1
Reviewer 1 Report
Critical modification should be conducted before considering it again for Sensors.
[1] The abstract too long and does not well convey the originality of current study. In particular, the main focus was devoted to apply for the health monitoring of dam system via previously developed signal process procedures, three Hjorth parameters, Cumulative sum, and Fuzzy logic so that the technical depth seems to be marginal for Sensors. You should reply them with a concrete manner for further review.
[2] Why the measured acoustic data can be successfully identified for your suggested signal processing? For example, if the acoustic data was judged by Fuzzy logic without applying for Hjorth parameters and Cumulative sum, what happen for the accuracy of health monitoring result?
[3] The introduction was not also well prepared to declare the main contents of this study; the chapter 2-4 should be merged into introduction section. In addition, several literature reviews were partially stated in several chapters. You should summarize them in introduction section.
[4] The main analytic tools, such as CUSUM algorism and PCA, were not well properly explained.
[5] Why you select alpha=0.01 for threshold value? If the threshold value was changed, the sensitivity of anomalies will be changed?
Author Response
Please see the attachment to Reviewer 1.

Reviewer 2 Report
Manuscript ID: sensors-1630249
Title: Structural Health Monitoring of Dams based on Acoustic Monitoring, Deep Neural Networks, Fuzzy Logic and a CUSUM Control Algorithm
Authors: Luan Carlos de Sena Monteiro Ozelim * , Lucas Parreira de Faria Borges , André Luis Brasil Cavalcante , Enzo Aldo Cunha Albuquerque , Mariana dos Santos Diniz , Manuelle Santos Góis , Katherin Rocio Cano Bezerra da Costa , Patrícia Figuereido de Sousa , Ana Paola do Nascimento Dantas , Rafael Mendes Jorge , Gabriela Rodriguez Moreira , Matheus Lima de Barros , Fernando Rodrigo de Aquino
In this manuscript, the authors presented the significance of Acoustic Monitoring and numerical methods for structural health monitoring of dams.
The Vajont dam failure costs more than 3000 lives. From history, there are many identical references. Therefore, to avoid any catastrophic failure, it is essential to monitor the structural health of dams.
The comments are as follows:
--- I did not find any novelty in this manuscript. The literature review is poorly presented. The problem statement is not discussed to meet the requirements of a technical paper.
---There is irrelevant information that does not carry any value to this manuscript.
--- The references are poorly presented. For example, on Page # 5, one reference was used more than nine times.
Structural health monitoring is a complicated procedure. Any algorithms to train experimental or non-destructive tests require a specific flow diagram which is missing. It is challenging to find the head and tail of this manuscript.
Further comments:
The summary of the critical issues are as follows:
--- The literature review is not significant to confine the originality. The authors must mention the significance of the statement of the problem and how it was addressed through the methodology presented in this manuscript. Also , the authors must address the novelty of the work.
--- The details of acoustic monitoring methods adopted for the study domain are absent in this manuscript. Other non-destructive tests, e.g., ground-penetrating radar or identical, are available in the literature for structural health monitoring. However, the authors did not address why acoustic monitoring is beneficial for the study. There are many conditions: dam type, geometry, geology, the purpose of the dams etc., which are crucial to determine the non-destructive tests and observational monitoring using instruments, e.g., sensor, which one used here and why it was selected; this information is not addressed in the manuscript.
---The study domain's geometry is not clearly presented in this manuscript.
--- Every algorithm has advantages and disadvantages considering the application domain. The limitation of algorithms and governing equations is missing in this manuscript, which could be interesting topics for future read.
It is highly recommended to consider English Editing services from colleagues or any professional to present the manuscript as a technical paper and avoid misleading information and presentation style.
Author Response
Please see the attachment to Reviewer 2.

Round 2
Reviewer 1 Report
Reviewer’s raised issues were relatively solved in the revised form. However, still some revisions should be conducted before publication.
[1] Still, the length of abstract was too long. ABSTACT should carry the main focus of your study and second important ones can be expressed in introduction section.
[2] In section 1, too any sub-sections were written so that it is recommended to shorten sentences concisely.
[3] Section 2 may be merged into section 3 by inserting them as sub-part to express the theoretical background regarding SM in section 3.
[4] All font type and size of figures should be matched.
Reviewer 2 Report
Accepted in the present form.
